# If Only You Could Catch Me—Catch Me If You Can: Monitoring Aphids in Protected Cucumber Cultivations by Means of Sticky Traps

**Christine Dieckhoff †** and **Rainer Meyhöfer \***

Institute for Horticulture and Production Systems, Section Phytomedicine, Applied Entomology, Gottfried Wilhelm Leibniz University Hannover, Herrenhäuser Str. 2, D-30419 Hannover, Germany; christine.dieckhoff@ltz.bwl.de
* Correspondence: meyhoefer@ipp.uni-hannover.de
† Current Address: Center for Agricultural Technology Augustenberg (LTZ), Neßlerstr. 25, D-76227 Karlsruhe, Germany.

**Abstract:** Aphids are important pests in many greenhouse and field crops. For plant protection, early detection of relevant species and reliable assessment of population development throughout the season is mandatory to address countermeasures in time. In practice, coloured sticky cards or pan traps are frequently used as monitoring tools, but as well as the flight activity of focal insects, many other factors influence reliable interpretation of trapping data. Since monitoring data have been more and more integrated into automated decision support systems, soundness of insect count data and interpretation of results needs to be reviewed in more detail. Therefore, we investigated the applicability of yellow sticky traps for monitoring of the cotton aphid, *Aphis gossypii* in greenhouse cucumber crop. In separate greenhouse chambers, we infested cucumber plants with *Aphis gossypii* and installed several yellow sticky traps. Insects were counted on the plants and sticky traps on a weekly basis and number of insects were correlated. Our results indicate mismatches between trap catches and aphid population density especially early in the season, which most likely is related to immigration of winged aphids into the greenhouse. The following population build-up of the cotton aphid *Aphis gossypii* on the cucumber plants correlated quite well with counts of alate cotton aphids on the sticky traps. In conclusion, trapping of winged aphids provides valuable information for integrated pest control in the greenhouse. Nevertheless, to avoid wrong interpretation the taxonomic identity of trapped aphids has to be confirmed at all times. Results are further discussed in the context of factors influencing aphid wing development and attraction to yellow sticky traps. Potential strategies to optimize aphid monitoring with coloured sticky traps are proposed.

**Keywords:** *Aphis gossypii*; *Cucumis sativus*; greenhouse crops; decision support systems; integrated pest management; action threshold; population development; yellow sticky traps



## 1. Introduction

Integrated pest management (IPM) is a comprehensive and sustainable management approach to protect crops against pests and diseases in order to avoid or minimise economic losses. Prompt and effective pest management decisions are at the core of successful integrated pest management. They require detailed knowledge of the pests and potential natural enemies as well as their respective biology and behaviour, adequate monitoring techniques, and efficient application of counter measures. Pest control measures can involve biological, cultural, or chemical approaches, or a combination thereof [1]. Management decisions should be based on the size of the pest population within a crop system relative to species-specific economic threshold values, above which economic losses are to be expected. In order to avoid economic damages, control measures need to be taken once a pest population reaches such a threshold. Advanced management decisions might also

consider the status of natural enemies already active in the crop, to predict pest population development in the coming weeks. Thus, continuous monitoring of a pest population within a crop system is crucial for successful IPM decisions [2,3].

There are several monitoring tools available to be used in horticultural crop settings. The most basic but also time-consuming method is a manual inspection of crop plants or plant parts for signs of infestations and assessment of pest densities. A more economical approach is the placement of traps within a crop to lure flying insects away from the host plant. Several types of traps are commercially available of which sticky traps are a popular and cost-effective option. A recommended trap density is one trap per 200 m$^2$ [4]. They are available in various colours, with yellow and blue being the ones most commonly recommended to growers as many pest insects have been shown to be attracted to either of those colours. For example, thrips, such as the Western Flower Thrips, are attracted to both yellow and blue wavelengths [5,6], while aphid and whitefly species respond well to yellow [7–10]. A study by Roach [11] found the cotton aphid, *Aphis gossypii*, to be significantly more attracted to yellow traps than another four colours tested. The insects' attractiveness to specific colours is not only based on the colour's wavelength as brightness, hue, reflectance intensity, contrast, and saturation may also play a role in a species' colour response [6,10,12–14]. Recently, van Tol et al. [15] found out that even the haziness of the glue used for insect trapping can play an important role. The attraction of insects towards colours, though, is usually more complicated than the response to colour characteristics alone. Studies have shown that several factors such as trap placement, environmental variables, and physiological state of the individuals also play a role in an individual's behavioural response. Straw et al. [16] found that traps placed in the upper half of spruce canopies caught significantly more spruce aphids, *Elatobium abietinum*, than traps placed in the lower half of the canopy. In a study by Byrne et al. [17], there was a significant difference between trap catches of two whitefly species with traps placed at three different heights in field crops. Gerling and Horowitz [9] found yellow traps to be most attractive to the tobacco whitefly, *Bemisia tabaci*, early in the season when individuals were looking for suitable host plants, i.e., after emerging from their overwintering habitats. Several aphid species, e.g., *Aphis fabae* and *Rhopalosiphum padi*, also showed different responses to trap colour stimuli depending on whether they were summer or autumn migrants [10,18,19]. With aphids, it has to be kept in mind that only the winged portion of the population can be caught with traps, and thus timing and trap placement is an important consideration in the monitoring of aphids and interpretation of trap catches.

For IPM decision making to be economic and sustainable, pest densities caught on traps need to be correlated with within-crop pest densities. Based on the resulting population density, estimates for optimal pest control measures can then be selected. Several studies have shown that trap catch levels are correlated well with population levels on the crop and can thus be used to assess pest population densities in relation to action thresholds. Böckmann et al. [20] showed that sticky trap catches of the greenhouse whitefly, *Trialeurodes vaporariorum*, were positively correlated with nymphal densities on tomato plants. The study also showed that such a correlation existed for the natural enemy, *Encarsia formosa*, released against *T. vaporariorum* in the same setting. Thus, the threshold concept may also be applied for natural enemies which then adds valuable information for the quality and impact of the biological control effort. Similar correlations between trap catches and resident pest populations were found for the western flower thrips, *Frankliniella occidentalis*, and the Asian psyllid, *Diaphorina citri* [13,21]. There is an increasing interest in using such correlations in the development of dynamic modelling approaches and automated decision support systems (DSS). Basically, such DSS are designed to help decision-makers, e.g., growers and extension specialists, in increasingly complex working environments by processing the data input based on set parameters and databanks in a timely manner. Decision-makers can then implement specific actions based on the DSS output. With the advancements in computer technology, the development of DSS is an ever evolving field of research and marketing. DSS for crop protection purposes have been designed since

the 1970s to aid growers in IPM decision-making processes in agricultural, horticultural and fruit crops [22–25]. In IPM decision-making, monitoring and subsequent pest control applications are time-sensitive and require swift action on part of the growers to prevent pest populations from reaching critical levels. Therefore, machine learning techniques and dynamic modelling approaches are increasingly implemented in modern-day DSS to forecast population dynamics and provide recommendations for courses of actions based on predicted pest densities in relation to set action thresholds [26–28].

In the current study, we focus on the melon or cotton aphid, *Aphis gossypii* Glover (Hemiptera: Aphididae), which is a polyphagous pest causing serious economic problems worldwide in agricultural as well as horticultural crops [29]. Plant damage is caused directly by plant feeding, through the transmission of various viruses, or indirectly through the excretion of honeydew promoting fungal growth and thus reducing photosynthetic activity [30,31]. The host range of *A. gossypii* contains numerous wild as well as cultivated plants in over 90 plant families within the Malvaceae, Solanaceae, Asteracea, and Cucurbitaceae, among others [29,30,32,33]. In temperate regions, *A. gossypii* is one of the major pests in cucumber crops, making it a serious economic threat [34,35]. In Germany, cucumbers, *Cucumis sativus* (Cucurbitaceae), are an economically important greenhouse and field vegetable crop. In 2022, the production area covered a total of 1854 ha, which was, in greenhouses, only second to tomato crops [36]. As is the case with most aphid species, *A. gossypii* has a high reproductive rate with both alate and apterous adults producing offspring, overlapping generations, and all developmental stages feeding on the plants. Winged individuals (alates) are primarily present during phases of migration such as during spring and autumn as well as being a result of declining host plant quality or crowding [37]. In addition, it has been shown that the presence of natural enemies has the potential to induce wing formation [38]. Wing polyphenism such as this is an important aspect of *A. gossypii*'s life cycle and contributes to its success as a pest species. It also makes monitoring of their populations with sticky traps a challenge as only a portion of the reproductive generation, i.e., alates, may be trapped and the proportion of winged aphids in the population is not constant over time.

Therefore, this study was designed to assess the potential correlation between sticky trap catches and on-plant populations of *Aphis gossypii* on cucumber, *Cucumis sativus*, in a greenhouse setting. The finding of such a correlation could then be used in future developments of automated monitoring and decision support systems.

## 2. Materials and Methods

### 2.1. Insect Material

*Aphis gossypii* Glover (Hemiptera, Aphididae) has been reared on *Cucumis sativus* cv. "Chinese Slangen" in self-made cages (approx. 30 × 30 × 40 cm) at the Leibniz Universität Hannover, Germany. Plants and cages were changed regularly to prevent mould and powdery mildew growth, respectively.

### 2.2. Plant Material

Cucumber plants, *Cucumis sativus* cv. "Chinese Slangen" (Weigelt Samen, Grolsheim, Germany) and cv. "Cum Laude RZ F1" (Rijk Zwaan Welver GmbH, Welver, Germany) were grown at the Institute of Horticulture and Production Systems, Leibniz University of Hannover, Germany. For the greenhouse experiments, only the variety "Cum Laude RZ F1" was used. Cucumber seeds were planted individually in growing medium (Substrat 1, Klasmann-Deilmann GmbH, Saterland, Germany) and fertilized regularly with a liquid fertilizer (0.2-% solution, WUXAL Top N, Manna, Nürnberg, Germany). When the plants reached the 3–4 leaf stage, they were transferred individually into 10-L pots with growing medium and approximately 50 g of a long-term fertilizer (Azet Tomato Fertilizer, W. Neudorff GmbH KG, Emmerthal, Germany).

### 2.3. Experimental Setup

The whole study was carried out in the experimental greenhouse facilities of Leibniz University, which are located in the centre of the city and surrounded by small experimental field plots as well as home gardens. All greenhouse compartments were 63 m² in size and computer-controlled ventilation windows along one side and in the roof allowed for temperature adjustment. Ventilation windows were not equipped with insect screens. In each of three greenhouse compartments, a total of 40 plants were arranged in four rows with ten plants each at a distance of approx. 0.5 m between pots within a row and 1.5–1.7 m between rows (Figure 1). Plants were trained on strings to a maximum height of 2.1 m at which point the main shoot was cut and two side shoots were allowed to grow downwards. Plant care was applied twice a week and consisted of removing both side shoots and young fruits up to the sixth node and only the side shoots from the seventh node through the top. Ripe fruits were harvested as necessary and plants were checked regularly for diseases such as powdery or downy mildew as well as infestations with spider mites. A drip irrigation system connected to a timer was set up to water the plants at regular intervals. Liquid fertilizer (0.2%-solution, WUXAL Top N) was given as needed. A polyethylene shading material (15%, 13 mm max. mesh size, Hermann Meyer KG, Langenau, Germany) was installed at a height of 3–3.5 m inside each cabin to prevent sunburn of the leaves. In addition, the outside walls of the greenhouse were coated with whitewash. The temperature inside the compartments was set to a minimum temperature of 18 ± 2 °C at night and 22 ± 2 °C during the day and it was monitored in 30 min intervals using data loggers (TinyTag Plus 2, Gemini Data Loggers (UK) Ltd. or HOBO Pendant®, Onset Computer Corporation, Bourne, MA, USA).

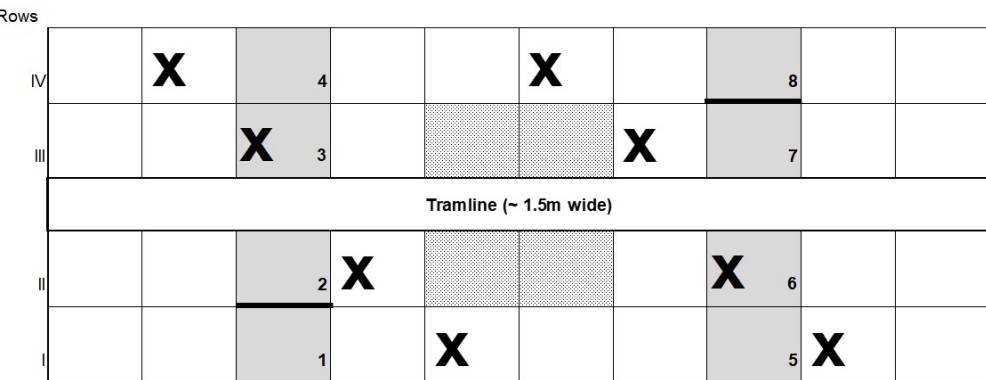

**Figure 1.** Schematic of the greenhouse compartment setup. **X** mark plants that were each infested with five alates and one apterous viviparous female *Aphis gossypii*. Grey fields mark the eight sampling plants. The bold black lines mark the location of the yellow sticky traps with trap 1 and 2 placed between sampling plants 1 and 2 and 7 and 8, respectively. The dotted area in the middle marks the space where releases of beneficial insects took place, if applicable.

After setting up the compartments, plants were given a couple of days to acclimatise before infesting a total of eight plants per compartment with five alate and one apterous viviparous female *A. gossypii* individuals each (Figure 1). In total 48 aphids were released in each compartment. Since biological control plays a major role in practice, *Aphidius colemani* Viereck (Hymenoptera: Braconidae) was released in two out of the three greenhouse compartments for aphid control. *Aphidius colemani* were ordered from Katz Biotech AG (Germany, Baruth). The remaining compartment served as untreated control, i.e., without release of natural enemies. The developing aphid population was monitored weekly by counting the insects on nine randomly selected leaves per plant (3 leaves each in the top, middle, and bottom third of the plant) of a total of eight sampling plants (Figure 1). If a plant did not have nine leaves then all developed and healthy leaves were included in the sampling. Sampling was performed in a non-destructive manner and the following

developmental stages were taken into account: 3rd and 4th nymphal stage, adult stages with and without wings (alates and apterae, respectively), and parasitized aphids (mummies that developed due to a minor natural infestation by parasitoids). For the statistical analyses, the two nymphal stages were combined as differentiation between the two stages was not reliable under the chosen non-destructive sampling procedure.

Two yellow sticky traps (1 trap/20 m$^2$; IVOG$^®$ Blanco Gelb, 10 × 25 cm) per compartment were installed one week after aphid introduction. Traps were placed just above the top height of the plants, adjusted with plant growth, and exchanged weekly, just prior to the weekly insect counts. The experiment was repeated twice in 2018: for the first planting, all data sampling was conducted between calendar week 25 and 32 (22 June to 21 August 2018) and for the second planting between week 39 and 45 (26 September to 7 November 2018). In both runs of the experiment, an infestation of spider mites occurred, necessitating the application of Floramite (Bifenazate 240 g/L) in calendar week 30, which caused all insect populations to crash in the weeks following. Between the first and second planting, the compartments as well as the equipment were disinfected using MennoFlorades (MENNO Chemie-Vertrieb GmbH, Norderstedt, Germany). During the second planting, no chemical pesticides against spider mites were applied and a minor case of powdery mildew occurred starting in week 40. All dead or diseased leaves were removed. Starting in calendar week 27 and 43, respectively, parasitized aphids (mummies) were observed in the control treatment of each planting due to a natural invasion of *Aphidius colemani* (Hymenoptera: Braconidae). Species identities of all protagonists was frequently verified by morphological characters.

### 2.4. Statistical Analyses

Both plantings were infested by spider mites which negatively affected overall plant health. Therefore, for the analyses of the first planting, insect counts for weeks 31 and 32 were excluded from the analyses as the aphid population abruptly dropped due to an application of an acaricide in calendar week 30. During the second planting, no miticides were applied but sampling data for calendar weeks 44 and 45 were excluded from the analyses as the heavy spider mite infestation interfered negatively with the plant health, and thus with the insect population dynamics.

Data were analysed using R (version 3.6.1) and RStudio (version 1.1.463). Linear mixed models were applied using the *lmer* function from the lmerTest package. Least square means were computed if applicable using the *emmeans* function of the emmeans package in R. Insect development over time was analysed by applying a linear mixed model with insect counts per plant as the response variable, calendar week as fixed effect and sampling plants and temperature as random effects. Calendar week and sampling plants were coded as factors in R. The Satterthwaite method was applied to the REML-fitted models. The response variables were log(x + 1) transformed. The relationship between mean counts of insect stage per plant and trap counts of winged individuals was analysed using linear mixed models. Data for traps were analysed on a per trap basis and correlated with average insect counts per plant of the 4 plants closest to each trap. That is, trap 1 was correlated with mean values of sampling plants 1, 2, 5, and 6 and trap 2 was correlated with mean values of sampling plants 3, 4, 7, and 8 (Figure 1). Models were fitted with mean counts for insects per plant (log(x + 1) transformed) as response variable, mean counts per trap as fixed effect and trap as random effect. Response variables used were mean number of alates per plant, mean total number of aphids per plant, and mean total number of aphids per plant of the week prior. Trap was coded as factor in R. Satterthwaite's method was applied to the REML-fitted models.

## 3. Results

### 3.1. Temperatures over Time

Temperatures over time were significantly different between the two plantings in both the controls ($F_{1,86}$ = 120.5, $p < 0.0001$) and the two treatment compartments ($F_{1,170}$ = 7.9706, $p = 0.005$). The average temperature ranged from 24.74 ± 0.24 °C to 29.71 ± 0.35 °C

during the first and from $23.15 \pm 0.15$ °C to $24.43 \pm 0.20$ °C during the second planting (Figure 2). In addition, temperatures between the two treatment compartments with release of *Aphidius colemani* were significantly different from each other within the first planting but not the second one (first planting: $F_{1,94} = 7.9199$, $p = 0.005$; second planting: $F_{1,76} = 1.8264$, $p = 0.1806$) (Figure 2C,D).

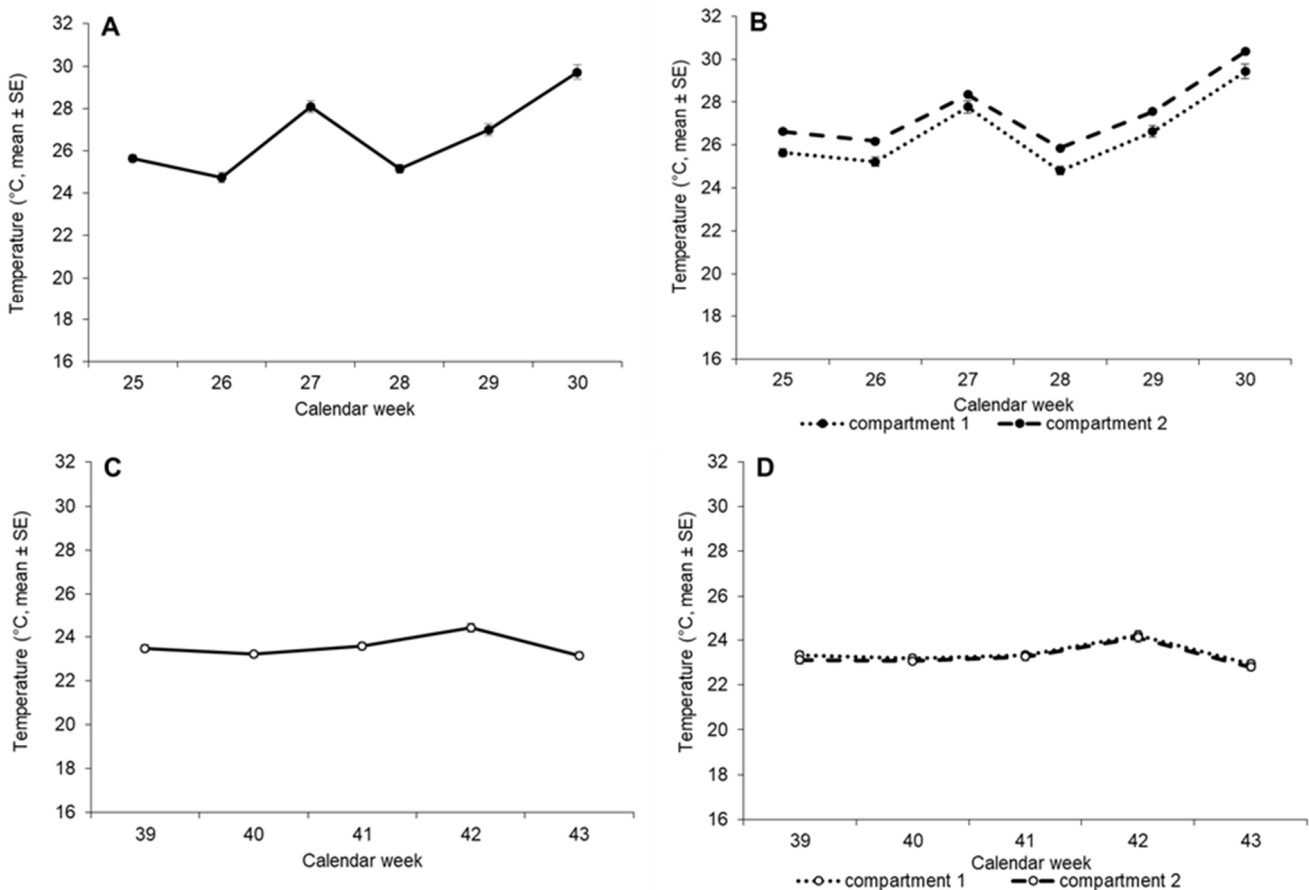

**Figure 2.** Temperatures over time in the compartments of the first (closed circles, upper graphs) and second (open circles, lower graphs) plantings: (**A**) control of the first planting (solid line), (**B**) compartment 1 (dotted line) and 2 (dashed line) of the first planting, (**C**) control of the second planting (solid line), and (**D**) compartment 1 (dotted line) and 2 (dashed line) of the second planting. Shown are average temperatures in °Celsius ($\pm$ SEM) per calendar week.

*3.2. First Planting—Insect Counts over Time*

In the control treatment of the first planting, the average counts of alates per plant and alates caught per trap changed significantly over time ($F_{5,40} = 8.3595$, $p < 0.0001$ and $F_{5,10} = 15.627$, $p < 0.001$, respectively). The average number of winged individuals per plant increased from calendar week 25 through 28 and declined afterwards, while on average most alates were caught on the traps in the first week of sampling (Figure 3A). Counts of alates per trap were not significantly different between traps 1 and 2 ($F_{1,5} = 5.2269$, $p = 0.07098$). Trap catches ranged from 1 to 30 winged aphids on trap 1 and from five to 22 aphids on trap 2, with individuals caught on either trap declining over time. While per-plant densities of aphids tended to be higher on plants associated with trap 2, there was no significant difference between the average per-plant densities of aphids on plants associated with each trap ($F_{1,5} = 2.4084$, $p = 0.1814$). Thus, trap placement within the greenhouse did not influence the correlation between plant and trap counts in the controls of the first planting. The average number of adult wingless aphids (apterae) per plant increased significantly over time with a maximum of $56.75 \pm 24.54$ individuals per plant in

week 29 ($F_{5,40}$ = 6.789, $p < 0.001$). The average number of nymphs per plant also increased significantly over time and reached a maximum average of 580.88 $\pm$ 256.99 individuals per plant in week 29 ($F_{5,40}$ = 8.588, $p < 0.0001$) (Figure 3B).

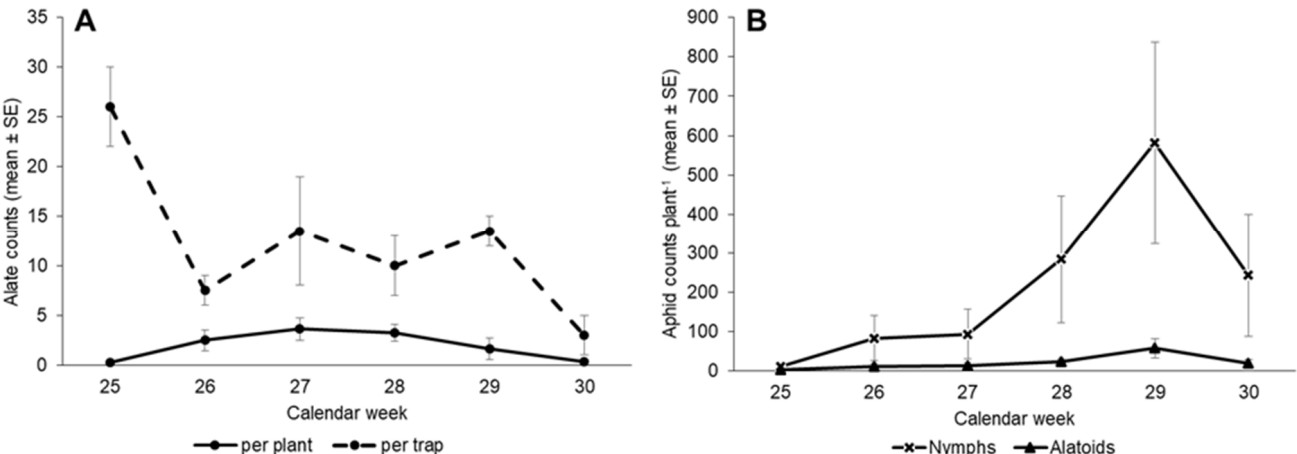

**Figure 3.** Aphid counts over time per plant or trap in the control compartment of the first planting. (**A**) alates per plant (solid line) and per trap (dashed line), (**B**) nymphs (solid line, cross) and apterae (solid line, triangle) per plant over time. Shown are average counts ($\pm$SE) per plant or trap and week, respectively.

In the two treatment compartments of the first planting, the average counts of alates per plant and alates caught per trap changed significantly over time ($F_{5,88}$ = 30.6993, $p < 0.0001$ and $F_{5,20}$ = 43.1664, $p < 0.001$, respectively). There was also a significant difference in trap counts between the two compartments over time ($F_{5,20}$ = 7.5387, $p < 0.001$). The average trap counts in compartment 1 fluctuated wildly between weekly counts, while in compartment 2 the number of insects increased steadily over time. On average, most alates were recorded on the traps in the first week of sampling (Figure 4A,C) the average number of wingless adult aphids per plant ranged from 2.88 $\pm$ 1.75 to 85.63 $\pm$ 52.65 in compartment 1 and from 2 $\pm$ 1.30 to 119.13 $\pm$ 60.48 in compartment 2. There were no significant differences between counts of trap 1 and 2 in either compartment ($F_{3,18}$ = 1.9779, $p = 0.1534$). In the two compartments, there was a significant increase in apterae counts through week 29 after which numbers declined ($F_{5,88}$ = 13.44, $p < 0.0001$). There was no significant difference between the two compartments ($F_{1,88}$ = 0.6397, $p = 0.426$). The average number of nymphs per plant significantly increased over time ranging from 16.13 $\pm$ 11 to 1154.63 $\pm$ 953.39 and from 4.63 $\pm$ 3.82 to 2860.38 $\pm$ 1791.37 in compartments 1 and 2, respectively ($F_{5,88}$ = 22.5131, $p < 0.0001$; Figure 4B,D). Nymphal counts per plant were not significantly different between the two compartments ($F_{1,88}$ = 1.6474, $p = 0.2027$; Figure 4B,D). The mean number of parasitized aphids (mummies) per plant increased significantly over time, again with no significant difference between the two compartments ($F_{5,11}$ = 18.2852 $p < 0.0001$ and $F_{1,11}$ = 4.6776, $p = 0.0527$, respectively). The average number of mummies per plant peaked at 10.5 $\pm$ 6.04 and 41 $\pm$ 20.79, respectively, in compartments 1 and 2.

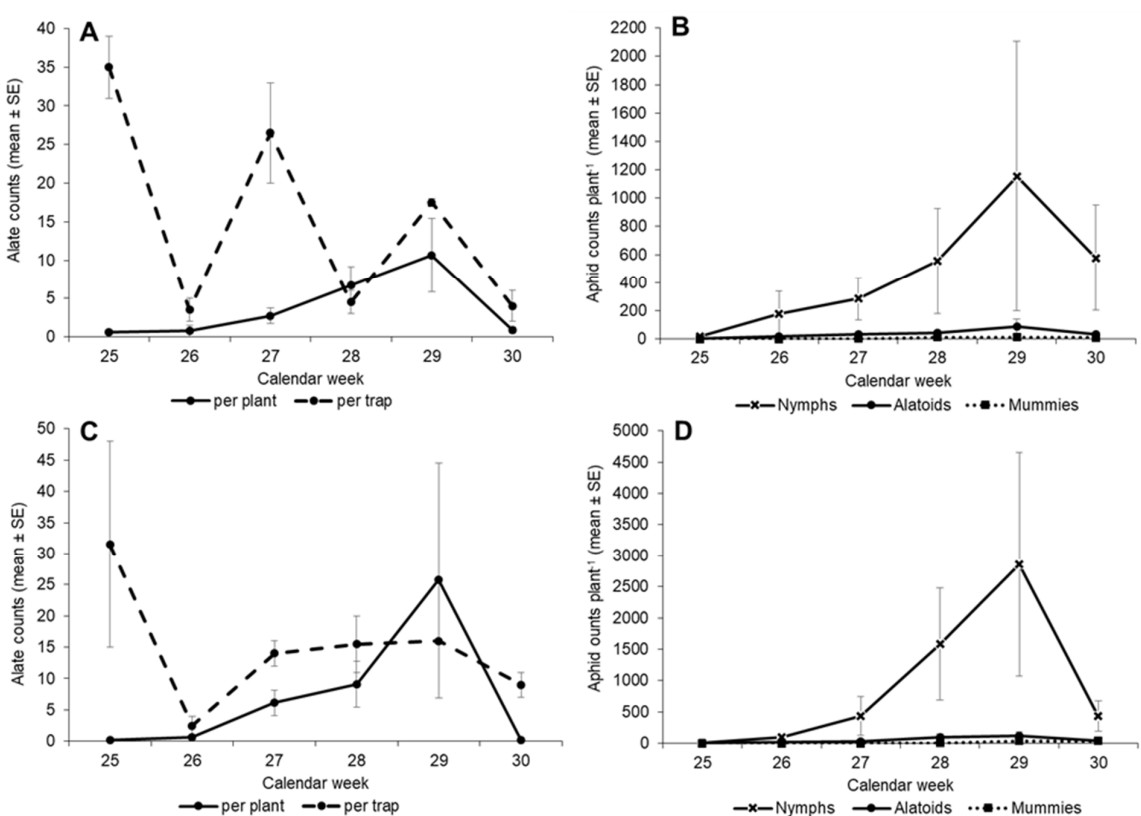

**Figure 4.** Aphid counts over time per plant or trap in the two treatment compartments of the first planting. (**A**) alates per plant (solid line) and per trap (dashed line) in compartment 1, (**B**) nymphs (solid line, cross), apterae (solid line, triangle), and parasitized aphids (mummies, dotted line, square) per plant in compartment 1, (**C**) alates per plant (solid line) and per trap (dashed line) in compartment 2, (**D**) nymphs (solid line, cross), apterae (solid line, triangle), and parasitized aphids (mummies, dotted line, square) per plant in compartment 2. Shown are average counts (±SE) per plant or trap and week, respectively.

*3.3. Second Planting—Insect Counts over Time*

In the control treatment of the second planting, mean number of alates per plant and trap had been changed significantly over time with number of adult winged aphids on plants increasing from week 40 through 43 and most individuals being caught per trap in weeks 42 and 43 ($F_{4,32}$ = 25.175, $p$ < 0.0001 and $F_{4,10}$ = 28.7785, $p$ < 0.0001, respectively) (Figure 5A). Average trap catches of winged aphids were significantly different between trap 1 and 2 ($F_{1,10}$ = 7.9239, $p$ = 0.01832) with trap 2 catching more alates than trap 1 in weeks 41 through 43. Throughout the sampling period (weeks 39 through 43) between 0 and 13 winged aphids were caught on trap 1 and between 0 and 20 alates on trap 2. The average number of alates per plant associated with trap 2 were 2.5-times and 4.2-times higher than plants associated with trap 1 in weeks 42 and 43 (interaction term: $F_{4,32}$ = 5.0621, $p$ = 0.002). Mean numbers per plant of both wingless aphids and nymphs increased significantly over time ($F_{4,32}$ = 20.358, $p$ < 0.0001 and $F_{4,32}$ = 34.367, $p$ < 0.0001, respectively). The average counts per plant peaked at 1937.38 ± 609.83 for nymphs and 133.63 ± 51.35 for apterae (Figure 5B).

In the treatments of the second planting, alates increased significantly over time per plant ($F_{4,70}$ = 27.791, $p$ < 0.0001) and counts were significantly different between the two compartments over time ($F_{4,70}$ = 16.627, $p$ < 0.0001). In compartment 1, the mean number of alates increased steadily over time while in compartment 2, the counts increased from week 39 to 40 and then decreased in the following weeks (Figure 6A,C). The average number of alates caught per trap also increased over time ($F_{4,16}$ = 13.896, $p$ < 0.0001) and were signifi-

cantly different between the two compartments ($F_{4,16} = 7.5997$, $p < 0.01$) (Figure 6A,C) and between the two traps by compartment ($F_{3,15} = 16.073$, $p < 0.0001$, not shown). Significantly more alates were caught by trap 1 in compartment 2 than by the other traps. The average number of wingless adult aphids per plant ranged from $4.13 \pm 2.59$ to $232.13 \pm 60.81$ in compartment 1 and from $49.25 \pm 28.13$ to $222.13 \pm 43.81$ in compartment 2. In either compartment there was a significant change over time through weeks 42 and 41, respectively, and after which time the average plant counts declined ($F_{4,70} = 42.896$, $p < 0.0001$). There was also a significant difference between the two compartments ($F_{1,70} = 51.792$, $p < 0.0001$) and the increase in aphid counts per plant over time was significantly different between the two compartments ($F_{4,70} = 15.154$, $p < 0.0001$). The average number of nymphs per plant significantly increased over time ranging from $9.63 \pm 5.15$ to $2470 \pm 665.22$ and from $313.13 \pm 219.15$ to $1490.63 \pm 397.91$ in compartments 1 and 2, respectively ($F_{4,70} = 47.663$, $p < 0.0001$; Figure 6). Nymphal counts per plant were significantly different between compartments and between compartments over time ($F_{1,70} = 48.931$, $p < 0.0001$; interaction term: $F_{4,70} = 14.805$, $p < 0.0001$). The mean number of parasitized aphids per plant increased significantly over time as well ($F_{4,10} = 10.84$, $p < 0.001$) but there was no difference between the two compartments. Average number of mummies per plant peaked at $111.71 \pm 19.65$ and $195.14 \pm 83.9$ in compartments 1 and 2, respectively (Figure 6B,D).

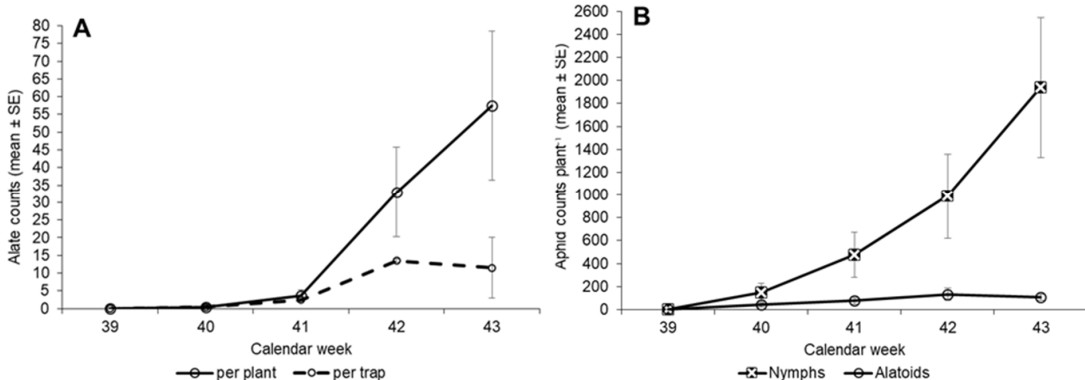

**Figure 5.** Aphid counts over time per plant or trap in the control treatment of the second planting. (**A**) alates per plant (solid line) and per trap (dashed line), (**B**) nymphs (solid line, cross) and apterae (solid line, triangle) per plant over time. Shown are average counts ($\pm$SE) per plant or trap and week, respectively.

*3.4. First and Second Planting—Correlation between Plant and Trap Counts in the Control Treatments*

In the first planting, there were no significant correlations between any of the response variables included in the models and the number of alates caught per trap (Table 1). In the second planting, there were significant correlations between all response variables tested (Table 1) and the number of alates caught per trap (Figure 7). Aphid plant and trap counts, specifically from trap 2 in weeks 42 and 43 had a strong influence on the correlation (Figure 7).

**Table 1.** Results for linear mixed models fitted for several response variables for the controls of the first and second planting. Shown are the regression equations, SEM (slopes), $r^2$, df, F and *p*-values, respectively.

|  | Response Variables | Regression Equation | Slope SEM | $r^2$ * | df | F | Pf > F |
|---|---|---|---|---|---|---|---|
| First planting | Number of nymphs previous week | $Y = 7.0237 - 1.3063X$ | 0.7461 | 0.24 | 10 | 3.07 | 0.1105 |
|  | Total number of aphids previous week | $Y = 6.9135 - 1.1794X$ | 0.7162 | 0.22 | 10 | 2.71 | 0.1306 |
|  | Number of alates same week | $Y = 0.72537 + 0.07919X$ | 0.26859 | 0.01 | 10 | 0.09 | 0.7742 |
|  | Total number of aphids same week | $Y = 4.8175 - 0.08067X$ | 0.65671 | 0.08 | 10 | 0.02 | 0.9047 |

**Table 1.** *Cont.*

| | Response Variables | Regression Equation | Slope SEM | $r^2$ * | df | F | Pf > F |
|---|---|---|---|---|---|---|---|
| Second planting | Number of nymphs previous week | Y = 1.7254 + 1.8992X | 0.3482 | 0.79 | 7 | 29.75 | <0.001 |
| | Total number of aphids previous week | Y = 2.1119 + 1.8081X | 0.3257 | 0.79 | 7 | 30.83 | <0.0001 |
| | Number of alates same week | Y = 0.06445 + 1.34920X | 0.18160 | 0.86 | 8 | 55.20 | <0.0001 |
| | Total number of aphids same week | Y = 3.6553 + 1.4888X | 0.3635 | 0.65 | 8 | 16.77 | <0.01 |

* $r^2$ is the conditional $r^2$ interpreted as variance explained by entire mixed model including both fixed and random effects (r.squaredGLMM function in MuMln package).

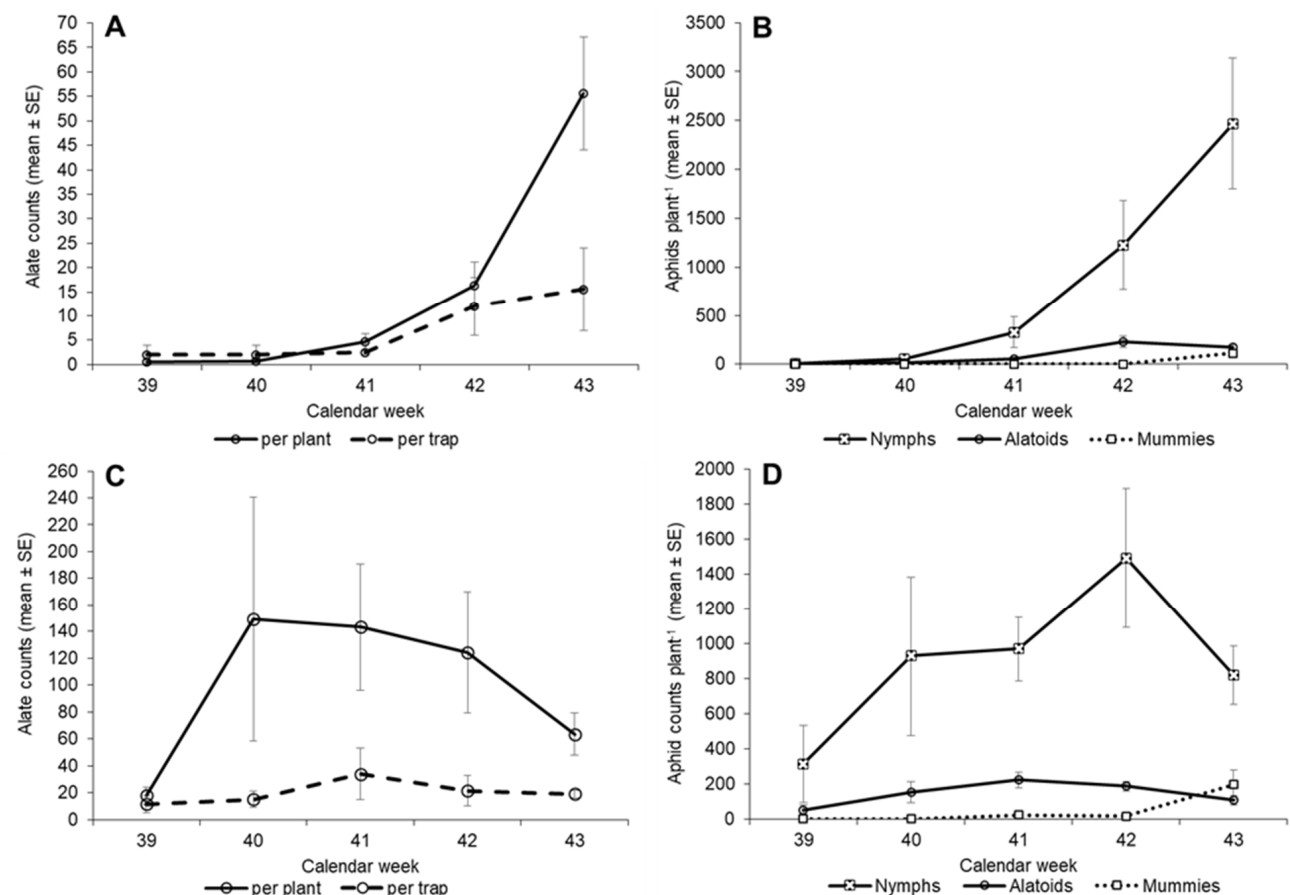

**Figure 6.** Aphid counts over time per plant or trap in the two treatment compartments of the second planting. (**A**) alates per plant (solid line) and per trap (dashed line) over time of compartment 1, (**B**) nymphs (solid line, cross), apterae (solid line, triangle), and parasitized aphids (mummies, dotted line, square) per plant over time in compartment 1, (**C**) alates per plant (solid line) and per trap (dashed line) over time of compartment 2, (**D**) nymphs (solid line, cross), apterae (solid line, triangle), and parasitized aphids (mummies, dotted line, square) per plant over time in compartment 2. Shown are average counts (±SE) per plant or trap and week, respectively.

*3.5. First and Second Planting—Correlation between Plant and Trap Counts in the Two Treatment Compartments*

There were no significant correlations between any of the response variables included in the models and the number of alates caught per trap in the treatments of the first planting (Table 2). In the treatments of the second planting, the mean trap catches of alates were correlated with the number of alates per plant of the same week and the total number of aphids per plant of the same week (Table 2, Figure 8). There were no significant correlations with the number of nymphs or aphids per plant of the previous week (Table 2).

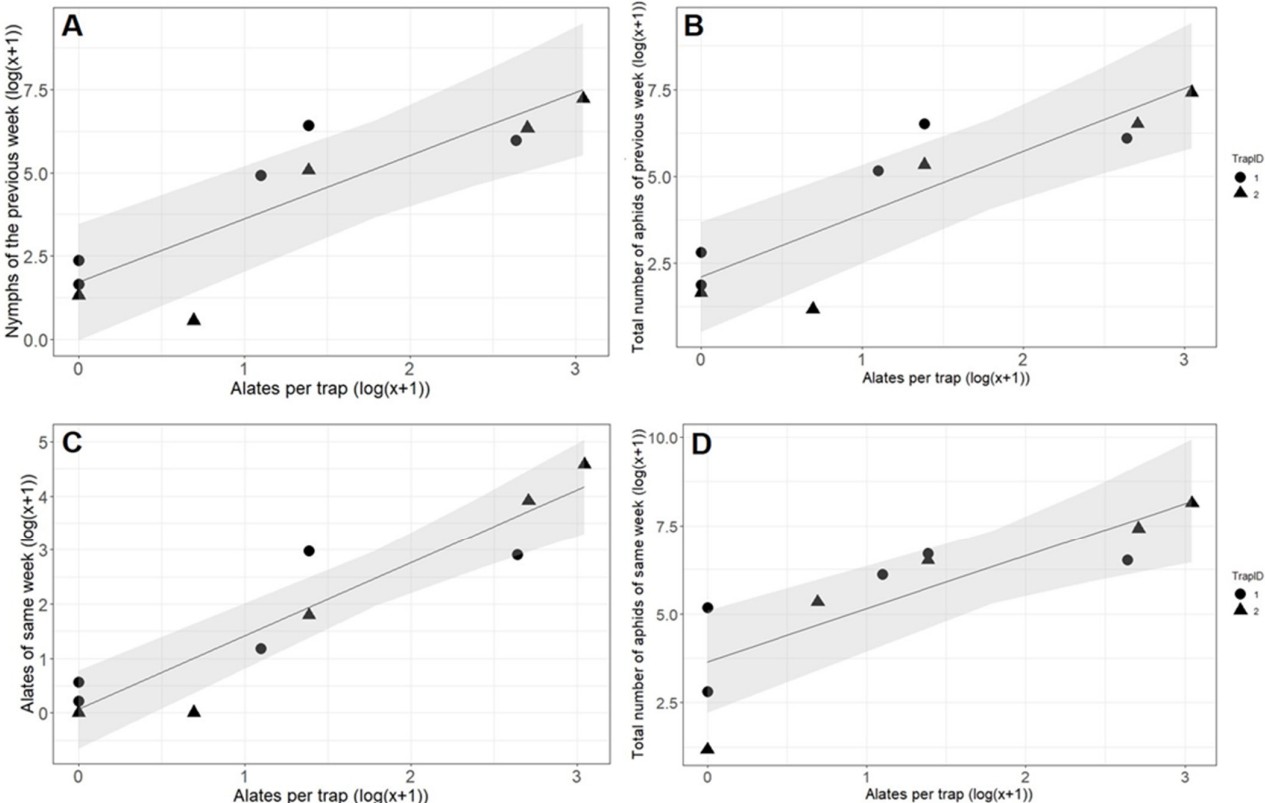

**Figure 7.** Correlation between average number of insect stages per plant and alates caught per trap per week in the control compartment of the second planting: (**A**) average number of nymphs per plant per week, (**B**) mean total number of aphids per plant per week of the previous week, (**C**) average number of alates per plant of the same week, and (**D**) mean total number of aphids per plant per week of the same week. The solid lines represent the mean estimates of each model and the grey areas delimitate the confidence intervals for the model estimates. Both x- and y-axes are log(x + 1) transformed. Statistical results are given in Table 1.

**Table 2.** Results for linear mixed models fitted for several response variables for the treatments of the first and second planting. Shown are the regression equations, SEM (slopes), $r^2$, df, F and *p*-values, respectively.

|  | Response Variables | Regression Equation | Slope SEM | $r^2$ * | df | F | Pf > F |
|---|---|---|---|---|---|---|---|
| First planting | Number of nymphs previous week | $Y = 3.5339 + 0.4374X$ | 0.5857 | 0.06 | 22 | 0.5577 | 0.4631 |
|  | Total number of aphids previous week | $Y = 3.8805 + 0.3831X$ | 0.5657 | 0.12 | 20 | 0.4585 | 0.5059 |
|  | Number of alates same week | $Y = 0.6319 + 0.2568X$ | 0.2508 | 0.04 | 22 | 1.0484 | 0.317 |
|  | Total number of aphids same week | $Y = 6.2326 - 0.3971X$ | 0.4592 | 0.29 | 20 | 0.7478 | 0.3975 |
| Second planting | Number of nymphs previous week | $Y = 4.5543 + 0.6371X$ | 0.3872 | 0.24 | 14 | 2.7078 | 0.1228 |
|  | Total number of aphids previous week | $Y = 4.7228 + 0.6557X$ | 0.3834 | 0.27 | 14 | 2.9259 | 0.1086 |
|  | Number of alates same week | $Y = -0.05012 + 1.40864X$ | 0.2271 | 0.87 | 18 | 38.465 | <0.0001 |
|  | Total number of aphids same week | $Y = 2.4525 + 1.6471X$ | 0.3208 | 0.83 | 18 | 26.359 | <0.0001 |

* $r^2$ is the conditional $r^2$ interpreted as variance explained by entire mixed model including both fixed and random effects (r.squaredGLMM function in MuMln package).

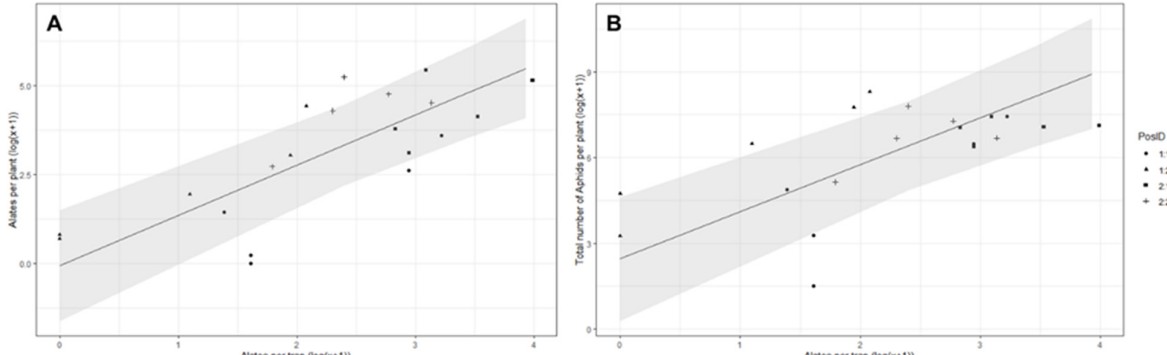

**Figure 8.** Correlation between average number of insect stages per plant and alates caught per trap per week in the two treatment compartments of the second planting: (**A**) average number of alates per plant per trap in the two treatment compartments, and (**B**) mean total number of aphids per plant per week of the previous week. The solid lines represent the mean estimates of each model and the grey areas delimitate the confidence intervals for the model estimates. Both x- and y-axes are log(x + 1) transformed, results are separated by trap per compartment. Statistical results are given in Table 2.

## 4. Discussion

In general, monitoring of aphids is directly related to perception and behavioural response of species to specific internal and external cues. Most important cues for aphid migration identified so far are visual [10] and volatile signals [39], including sex pheromones [39] as well as herbivore-induced volatiles [40]. Of general importance for all the different aphid morphs are visual cues, while semiochemicals are especially important for sexual reproduction and autumn migrants. Therefore, coloured targets are currently among the main tools for aphid monitoring in practice. Most often yellow sticky traps are used [41–43], but also yellow water traps [44,45] give good results. Moreover, visual response to colours is of major importance for spring migrants [46], while at the end of the season combinations with female sex pheromones allows detection of migrating males to winter hosts [47–49].

In the current study, we assessed the applicability of standard yellow sticky traps as a reliable monitoring tool for the cotton aphid, *Aphis gossypii*, in the context of decision support systems for IPM purposes in a greenhouse cucumber crop. There was a significant correlation between trap and on-plant aphid counts but only in the second of two plantings which was set up in the last quarter of the year 2018. In the first planting, no such correlation was found and, what is more important, the highest number of alates were trapped in the first week of sampling.

The results indicate that early in the growing season traps may only be a reliable monitoring tool when aphids emigrate from their overwintering hosts and start invading greenhouses in their search for suitable host plants. In temperate regions, *A. gossypii* overwinters as egg stage and commonly remains on the overwintering host to feed and build up a spring population before dispersing to a summer host [30,50]. Taking into account the initial number of alatae released per compartment and the slow build-up of winged individuals in the compartments over time, it is possible that the early peak in trap counts in the first planting (third week of June) was at least partially the result of immigrating individuals from the surrounding landscape. There were no insect screens covering the greenhouse windows to prevent insects from immigrating throughout the course of the experiment. In temperate regions, protected cultivation of cucumber crops is usually conducted in two to three sets per year [51,52]. Thus, if the observed early peaks represented immigrating alates, then pest control measures would be most successful if applied during the first set of cucumbers to prevent population build-ups at this early stage. On the other hand, the individuals caught on the traps in the first week of sampling may also have been those that were used to infest the plants with at the beginning of the experiment. Each first-week peak in the first planting was followed by a crash in trap

counts which would be consistent with a high proportion of the initial number of aphids used to infest the plants with being trapped. This would imply that the traps were more attractive than the host plants themselves which were of overall good quality at this point in time. The influence of time of the year in combination with trap colour on trap catches of aphids has been shown for several aphid species [10,18]. A study by A'Brook [18] found a significant correlation between trap colour and month in several species of aphids. *Aphis fabae,* for example, responded strongest to yellow traps in July compared to other trap colours. If this, however, was the main factor causing the observed high trap counts then, from an IPM point of view, this would be a false positive resulting in unnecessary and costly control measures against the perceived high numbers of aphids on the plants.

During the second planting, trap catches were significantly correlated with the average number of aphids on the plants and trap counts reflected the on-plant population build-up. The initial number of winged and wingless aphids used to artificially infest the experimental plants had been the same as during the first planting. The lack of trap catches in the beginning of the second planting (third week of September) would support an influence of season on the aphids' attraction towards the traps. The observed increase in trap catches over time during the second planting is more consistent with an increase in number of alates due to declining plant quality and/or overcrowding [30]. The presence of natural enemies in either one of the treatment compartments may have contributed slightly to the development of alatae. Mondor et al. [53] found that a higher predation risk resulted in an increase in winged *A. gossypii*. This was also found to be the case in the pea aphid, *Acyrthosiphon pisum*, when in the presence of predators [54]. However, it is considered unlikely that this was the case as the aphid population growth does not indicate a strong influence of either a predator or parasitoid. Observed parasitism caused by *Aphidius colemani*, the only parasitoid released in this experiment, was overall very low.

The internal state of the alatae may also have played a role in the attraction towards the sticky traps. At the beginning and the end of the season—i.e., during the peaks of the migrating population—most individuals were caught on the traps. Throughout the season, though, when host plants were in good or at least decent health, far less individuals were caught on the traps as was to be expected as apterae made up the bulk of the population. Similar behavioural patterns have been observed in other herbivorous insects.

Trap placement within the greenhouse compartments of the second planting had a significant influence on the outcome of the correlation regarding population densities within the crop. Thus, depending on the placement of a trap within a greenhouse and the area covered by each trap, the actual aphid density per plant might be underestimated and biocontrol measures might be taken too late to prevent economic damage within a crop. IPM decisions would then have to be based on a per-trap basis with each trap covering a pre-defined area within a greenhouse. In this experiment, two traps per compartment were set up with each trap covering about 20 $m^2$. This is considerably less area covered per trap than the recommendation of 1 trap per 200 $m^2$. It is in contrast to the findings of Böckmann et al. [20] which found that one trap per 170 $m^2$ was sufficient to reliably monitor *Trialeurodes vaporariorum* in a greenhouse tomato crop.

In this study, population growth itself was not negatively affected by high temperatures, especially during the first planting when mean temperatures were generally well above 26 °C. The abrupt decline in population densities that could be observed between the second to last and last week of observations in some of the treatments in either planting were rather the result of the poor plant quality (due to secondary pest infestations) than temperature conditions at the time. The temperatures during either planting were within the reported optimum developmental and reproductive temperature range for *A. gossypii.* Studies on various host plants, including cucumber, have shown that *A. gossypii* did not exhibit increased mortalities at temperatures above 25 °C, contrary to many other aphid species with lower optimum temperatures [30,55]. Optimal temperatures for population growth of *A. gossypii* on cucumber were found to be between 22.5 °C and 30 °C [56]. Life table studies of populations on cucumber [57] and cotton [58] reported fecundity of *A.*

*gossypii* to be highest at 30 °C and 25 °C, respectively, while development was fastest at 30 °C on either host plant. Satar et al. [56], though, found that daily fecundity rates of *A. gossypii* peaked at 25 °C. The differences in fecundity rates reported by Satar et al. [56] and Kocourek et al. [57] might be, at least partially, the result of the different cucumber varieties used in their experiments as experimental setups were otherwise comparable. An effect of host plant variety on life history parameters of *A. gossypii* was also found in a study by van Steenis and El-Khawass [55].

Finally, the experimental design used here may have had an influence on the observed trap catches. Traps may have not been placed at the most optimal height or place within the greenhouse. Traps were placed just at the top height of the cucumber plants instead of within the crop stand. This is the location commonly chosen by growers for placing these types of traps in greenhouse crops as such a placement does not interfere with plant care measures or harvesting activities. An effect of trap placement within the canopy was found for *Elatobium abietinum*, for example, where traps placed higher above ground caught more aphids compared to those placed at lower levels [16]. Hoelmer et al. [59] also found trap orientation relative to the crop to have an effect on an insect species' attraction towards the trap.

In conclusion, yellow sticky traps are a valuable tool for aphid monitoring in the greenhouse crops. On the one hand, early immigration of aphids can be detected when considering the species identity and might give first alerts for immediate pest control action. On the other hand, trap catches of winged adults increase only slightly during the growing season and correlate quite well with the actual aphid population density on the crop. Nevertheless, aphid counts on the monitoring trap in general are low and make it difficult to extract reliable action thresholds from the current study, but a follow up study by Grupe et al. [60] on sticky trap monitoring of the parasitoid foraging activity in the crop highlights the potential of indirect monitoring of wingless pest species and will contribute in the future substantially to threshold development. Finally, estimations and validation at larger scales in practice are necessary, and considering the trap densities in the crop also leaves room to improve the sensitivity of the monitoring system.

**Author Contributions:** Conceptualization, C.D. and R.M.; methodology, C.D. and R.M.; formal analysis, C.D.; investigation, C.D.; resources, R.M.; data curation, C.D.; writing—original draft preparation, C.D.; writing—review and editing, C.D. and R.M.; visualization, C.D.; supervision, R.M.; project administration, R.M.; funding acquisition, R.M. All authors have read and agreed to the published version of the manuscript.

**Funding:** The project was supported by funds of the Federal Ministry of Food and Agriculture (BMEL, FKZ: 2814903515) based on a decision of the Parliament of the Federal Republic of Germany via the Federal Office for Agriculture and Food (BLE) under the innovation support program.

**Institutional Review Board Statement:** Not applicable.

**Informed Consent Statement:** Not applicable.

**Data Availability Statement:** The data that support the research findings of this study is available at LUH Data Repository [61]: https://doi.org/10.25835/ami9gk33 (accessed on 8 May 2023).

**Acknowledgments:** We would like to thank Lisa Wittig for her help with insect rearing and counting and plant nursing.

**Conflicts of Interest:** The authors declare no conflict of interest.

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
