# Peer review of "If Only You Could Catch Me—Catch Me If You Can: Monitoring Aphids in Protected Cucumber Cultivations by Means of Sticky Traps"

_horticulturae, doi:10.3390/horticulturae9050571_

Round 1
Reviewer 1 Report
The manuscript by Dieckhoff and Meyhöfer describes a well-replicated greenhouse study on the applicability of yellow sticky traps in monitoring the cotton aphid populations in melon crops. Also, the manuscript is very well written. I could not find control details in the material method section. I want authors to include this information. Other than this I have minor comments below.
Line 38: remove of
Line 53: remove,” per area monitored”
Rephrase this this line, “his was also found to be the case in several aphid species, in addition to a significant difference in aphid trap catches between summer and autumn migrants, e.g., in Aphis fabae and Rhopalosiphum padi [10,18,19].”
Line 112: thus reducing photosynthetic activity [30; https://doi.org/10.3390/v12070773).
Just 231 ha in the whole country?
Line 137: Plants and cages were changed regularly to prevent mold and powdery mildew growth on cages and plants, respectively.
Line 150: What were the conditions (RH, Temp, L:D) in the greenhouse?
Line 199: control treatment? So out of three compartments one of control. Please clarify the control.
Line 297: changed significantly.
Line 479: A. gossypii
If aphids were immigrating from the outside, they could easily belong to different species. Did you find anything other than cotton aphids on traps or plants?
Line 19: with different aphid densities? Unless I missed something, each treatment compartment had 48 aphids.
Author Response
Dear Reviewer,
Thanks a lot for your helpful comments and suggestion. They substantially improved the manuscript and are all included in the revised version. Please find below the detailed point by point response. If you have further questions please let me know.
With kind regards
Rainer Meyhöfer
Detailed response:
I could not find control details in the material method section.
- Please see lines 176-178, more specific information about the untreated control was added.
Line 38: remove of
- done
Line 53: remove,” per area monitored”
- done
Rephrase this this line, “his was also found to be the case in several aphid species, in addition to a significant difference in aphid trap catches between summer and autumn migrants, e.g., in Aphis fabae and Rhopalosiphum padi [10,18,19].”
- done
Line 112: thus reducing photosynthetic activity [30; https://doi.org/10.3390/v12070773 ).
- done
Just 231 ha in the whole country?
- corrected
Line 137: Plants and cages were changed regularly to prevent mold and powdery mildew growth on cages and plants, respectively.
- done
Line 150: What were the conditions (RH, Temp, L:D) in the greenhouse?
- Temperature profiles are presented in figure 2. RH and light are strongly influenced by watering and sunlight and therefore not explicitly included.
Line 199: control treatment? So out of three compartments one of control. Please clarify the control.
- Please see lines 176-178, more specific information about the untreated control was added.
Line 297: changed significantly.
- done
Line 479: A. gossypii
- done
If aphids were immigrating from the outside, they could easily belong to different species. Did you find anything other than cotton aphids on traps or plants?
- Traps were checked frequently for aphid species identity. Other species then A. gossypii were rarely detected. Therefore, it is unlikely that they build up populations in the crop and/or compete with Aphis gossypii.
Line 19: with different aphid densities? Unless I missed something, each treatment compartment had 48 aphids.
- Yes you are right. The drag and drop error was corrected.
Reviewer 2 Report
The study is very interesting and provides useful information on using sticky traps as a crucial tool in the IPM module by establishing relationships with the various monitored parameters on greenhouse cucumber plants helping out to monitor and take timely action in managing the aphid population. The authors have nicely planned and executed the study. They have done the appropriate data analysis needed for such a study and also presented the results in a comprehensive manner to draw useful inferences.
There is no critical issue observed except for some minor errors noticed which authors must correct in the revision
L 11: Insert comma (,) before early
L 13: Insert comma (,) before colored
L 18: correct as ‘greenhouse cucumber crop’
L 18: insert a comma after cotton aphid,
L 19: greenhouse chamber seems a more appropriate word than cabin.
L 19: insert a comma before we infested
L 134: Use numbering order of subheads according to journal format throughout the manuscript.
L 392: Insert a comma after In general,
L 479: Itlicize A. gossypii
L 498: Place comma after conclusion,
L 498: comma after on the other hand,
L 521: References: please check carefully;
- please provide an abbreviated form of the journal name as it is missing in many references (5,7,9….);
- insert dot (.) after the abbreviated form of the journal (Ref. 6, 14, 16, 17…).
- Italicise the scientific name of inset/ crops (e.g. Ref.16)
- remove caps in references where not needed (e.g., 46)
Best
Author Response
Dear Reviewer,
Thanks a lot for your helpful comments and suggestion. They substantially improved the manuscript and are all included in the revised version. Please find below the detailed point by point response. If you have further questions please let me know.
With kind regards
Rainer Meyhöfer
Detailed response:
L 11: Insert comma (,) before early
- done
L 13: Insert comma (,) before colored
- done
L 18: correct as ‘greenhouse cucumber crop’
- done
L 18: insert a comma after cotton aphid,
- done
L 19: greenhouse chamber seems a more appropriate word than cabin.
- done
L 19: insert a comma before we infested
- done
L 134: Use numbering order of subheads according to journal format throughout the manuscript.
- done
L 392: Insert a comma after In general,
- done
L 479: Itlicize A. gossypii
- done
L 498: Place comma after conclusion,
- done
L 498: comma after on the other hand,
- done
L 521: References: please check carefully;
- done
- please provide an abbreviated form of the journal name as it is missing in many references (5,7,9….);
- done
- insert dot (.) after the abbreviated form of the journal (Ref. 6, 14, 16, 17…).
- done
- Italicise the scientific name of inset/ crops (e.g. Ref.16)
- done
- remove caps in references where not needed (e.g., 46)
- done
Reviewer 3 Report
(ISSN 2311-7524) horticulturae-2340043
If only you could catch me - Catch me if you can – Monitoring aphids in protected cucumber cultivations
The manuscript “If only you could catch me - Catch me if you can – Monitoring aphids in protected cucumber cultivations” describes the potential correlation between sticky trap catches and the population on plant of aphis in green house conditions. The outcome of the manuscript has relevance for the development of automated monitoring and decision making.
The highest value of the study was its completeness – two cultivars of Cucumis sativus, three greenhouses compartments, two yellow sticky traps and two repetitions. Taking into consideration the biology and life cycle of the aphis, the results are in general consistent.
The information is new and useful to monitoring aphis population in the context of decision support systems for IPM purposes in a greenhouse cucumber crop.
In general, the experiments are well designed, sized and repeated, to explain the applicability of standard yellow sticky traps as a reliable monitoring tool for the cotton aphid, Aphis gossypii. The data support the conclusions of the authors, and the results are discussed appropriately.
Author Response
Dear Reviewer,
Thanks a lot for your report and positive feedback! If you have further questions please let me know.
With kind regards
Rainer Meyhöfer
Reviewer 4 Report
The objective of the revised manuscript is to see the applicability of the use of yellow sticky traps to monitor the population dynamics of Aphis gossypii in greenhouse cucumber crops, with a view to its possible applicability in decision support systems.
It currently presents a several substantive problems for publication, so it is suggested that the authors modify them according to the following comments.
Comments to the Title
The authors should review the title, since it would be much better if only the final part "Monitoring aphids in protected cucumber cultivations" were included, because it is better suited to the development of the work: monitoring of the population dynamics of Aphis gossypii in cucumber and monitoring of the winged forms by means of sticky traps.
Comments to the introduction
The authors include a series of examples on trapping whiteflies, thrips and Diaphorina citri, aren't there examples that illustrate these same arguments with aphids?
Comments to the material and methods section
There are some deficiencies in the section that prevent a correct understanding of some of the results presented and their subsequent discussion:
1) It is necessary to clearly explain if the experiment was carried out in conventional greenhouses in which a normal production of cucumbers was already carried out, or if they were used only for this experiment.
2) At this point, the location of greenhouses and the environment that surrounds them should be indicated (if they are located in the middle of the countryside or in the city, if they are surrounded by weeds, forests...)
3) The dimensions of the greenhouses in which the experiments were carried out should be clearly indicated, as well as the dimensions of the breathing windows, also indicating whether or not they were protected from the entry of insects, because this aspect is not clear.
4) The position of the breathing windows should be indicated in relation to the location of the sticky traps.
5) It is not clear why the authors decided to include 5 winged and one ¿alatoid? in each plant. It should be indicated if any specific methodology is followed. From what can be understood from reading the manuscript, it seems that what the authors refer to with the word alatoid is an apterous viviparous female, which is clearly an error and must be corrected. This aspect should be clarified in all text.
6) It is also not very clear why they decided to count only third and fourth instar nymphs, winged and ¿alatoid nymphs?. What happens to the first and second instar of nymphs? Why don't they decide to differentiate between third and fourth instar alatoid nymphs from those same instar nymphs of the apterae? It is all very confusing and should be clarified because they decide to count some shapes and not others.
7) Apart from that, I think the authors are confusing alatoid with apterous viviparous females.
8) It seems from reading the manuscript that parasitoids of the species Aphidius colemani are incorporated into the greenhouses, but if so, it should be indicated in the Material and methods section where they come from, just as a section for aphids and plants has been included. By the way, parasitism is not mentioned in the abstract and should be indicated.
9) It is not clear why the controls exist. Is it obvious that the same aphids were included in the control plants as in the two greenhouses in which the experiments were carried out, but that Aphidius colemani parasitoids were not introduced into the controls, but rather they entered alone? Thus, these aspects should be clarified. In any case, how do the authors know that the species that parasitize the aphids in the controls are Aphidius colemani? It could be other species of parasitoids and hyperparasites
10) The authors should indicate how they identify the aphids that are caught in the sticky traps. Because some species of aphids of the genus Aphis can be confused with Aphis gossypii. This aspect is important, especially if the greenhouses allowed aphids to enter and exit them.
11) The same happens with parasitoids. The authors assume that all aphid mummies are caused by Aphidius colemani, but other species of parasitoids and hyperparasitoids could be parasitizing colonies inside the greenhouse. Especially if the greenhouses had open ventilation windows. They should indicate the coloration of the mummies.
12) Do not parasitoids or hyperparasitoids fall into the yellow traps or have they not been counted?
13) It would be convenient to explain in detail the state of development of the plants when the aphids are placed on the plants and to know how much time passes from when they are left on the plants until the traps are removed for the first time at week 25. This could clearly explain the massive capture of winged aphids in the first planting.
Comments to the results
In figure 2 the controls could be linked to the results in compartments 1 and 2
In figures 3, 4 and 5 it would be better if the three forms studied on the plant (apterae, alatae and nymphs) and the winged ones captured in the trap were included in a single graph. In such a way that there would only be one figure that included experiments 1, 2 and control for the first experiment and another figure with the same structure for the second.
Comments to the discussion
It is going too far to think that the development of wings is due to the presence of parasitism, especially when the data on the release of parasitoids have not been discussed (or are not known).
The authors refer to general issues of the life cycle of Aphis gossypii, but it would be necessary to know how it behaves in the area in which the experiments are being carried out, because at the time when the tests are being carried out, Aphis gossypii may be producing winged for many months and present several peaks of production of winged or a constant production. Are there no studies on this aspect in those areas? There is also no data on catches with previous traps?
For other comments and small errors, see also the attached corrected manuscript

Author Response
Dear Reviewer,
Thanks a lot for your helpful comments and suggestion. They substantially improved the manuscript and are all included in the revised version. Please find below the detailed point by point response. If you have further questions, please let me know.
With kind regards
Rainer Meyhöfer
Detailed response:
Comments to the Title
The authors should review the title, since it would be much better if only the final part "Monitoring aphids in protected cucumber cultivations" were included, because it is better suited to the development of the work: monitoring of the population dynamics of Aphis gossypii in cucumber and monitoring of the winged forms by means of sticky traps.
- thanks for the comment, we partly rephrased the title to highlight the importance of sticky traps in the current study.
Comments to the introduction
The authors include a series of examples on trapping whiteflies, thrips and Diaphorina citri, aren't there examples that illustrate these same arguments with aphids?
- Yes, you are right. There is a large body of literature on aphids available. But unfortunately literature on some facts is rare. Therefore, we included examples with other important greenhouse pests.
Comments to the material and methods section
There are some deficiencies in the section that prevent a correct understanding of some of the results presented and their subsequent discussion:
1) It is necessary to clearly explain if the experiment was carried out in conventional greenhouses in which a normal production of cucumbers was already carried out, or if they were used only for this experiment.
- done, information was added see line 150-154
2) At this point, the location of greenhouses and the environment that surrounds them should be indicated (if they are located in the middle of the countryside or in the city, if they are surrounded by weeds, forests...)
- done, information was added see line 150-154
3) The dimensions of the greenhouses in which the experiments were carried out should be clearly indicated, as well as the dimensions of the breathing windows, also indicating whether or not they were protected from the entry of insects, because this aspect is not clear.
- done, information was added see line 150-154
4) The position of the breathing windows should be indicated in relation to the location of the sticky traps.
- done, information was added see line 150-154
5) It is not clear why the authors decided to include 5 winged and one ¿alatoid? in each plant. It should be indicated if any specific methodology is followed. From what can be understood from reading the manuscript, it seems that what the authors refer to with the word alatoid is an apterous viviparous female, which is clearly an error and must be corrected. This aspect should be clarified in all text.
- the decision to include a small number of winged and wingless aphids was based on own experience and preliminary results. Furthermore, the error was corrected throughout the manuscript.
6) It is also not very clear why they decided to count only third and fourth instar nymphs, winged and ¿alatoid nymphs?. What happens to the first and second instar of nymphs? Why don't they decide to differentiate between third and fourth instar alatoid nymphs from those same instar nymphs of the apterae? It is all very confusing and should be clarified because they decide to count some shapes and not others.
- initially counting of aphids should give us a good estimate of population density of aphids on the plant. To minimise workload and staff training we therefore decided not to count the smaller developmental stages nor to distinguish between third and fourth instars. Looking at the results this probably should have been done and will be considered in follow up experiments. Nevertheless the results are quite clear and also the population development is reflected by the counting well.
7) Apart from that, I think the authors are confusing alatoid with apterous viviparous females.
- done, the error was corrected throughout the manuscript
8) It seems from reading the manuscript that parasitoids of the species Aphidius colemani are incorporated into the greenhouses, but if so, it should be indicated in the Material and methods section where they come from, just as a section for aphids and plants has been included. By the way, parasitism is not mentioned in the abstract and should be indicated.
- yes, the intention was to grow the cucumber crop in the greenhouse as close as possible to commercial crops in greenhouses. Therefore, we decided to include the release of parasitic wasps as part biological plant protection and leave one compartment untreated for the control. Unfortunately, parasitism rate was very low during both plantings. This was the reason why we finally did not consider the role of parasitoids for aphid monitoring in this study. Specific data on parasitoids and their role in aphid monitoring were studied in a follow up experiment and will be published elsewhere. (see lines 181-183 and 514-517)
Moreover, the focus of the current study was on aphid monitoring only and therefore we don’t see the need to include parasitism in the abstract.
9) It is not clear why the controls exist. Is it obvious that the same aphids were included in the control plants as in the two greenhouses in which the experiments were carried out, but that Aphidius colemani parasitoids were not introduced into the controls, but rather they entered alone? Thus, these aspects should be clarified. In any case, how do the authors know that the species that parasitize the aphids in the controls are Aphidius colemani? It could be other species of parasitoids and hyperparasites
- see response above. The unexpected emigration of parasitoids in the control treatment as well as the general low parasitism rate made it impossible to consider the specific role of parasitism, i.e. comparison with the control, in this study. Nevertheless, parasitoids on traps as well as parasitoids emerging from mummies were frequently determined to verify Aphidius colemani identity. (see line 208)
10) The authors should indicate how they identify the aphids that are caught in the sticky traps. Because some species of aphids of the genus Aphis can be confused with Aphis gossypii. This aspect is important, especially if the greenhouses allowed aphids to enter and exit them.
- see response above. Species identities of all protagonists was frequently verified by morphological characters. (see line 208)
11) The same happens with parasitoids. The authors assume that all aphid mummies are caused by Aphidius colemani, but other species of parasitoids and hyperparasitoids could be parasitizing colonies inside the greenhouse. Especially if the greenhouses had open ventilation windows. They should indicate the coloration of the mummies.
- see response above. Activity of other aphid parasitoid and hyperparasitoid species is unlikely but cannot be fully excluded. Since parasitism was low and irrelevant for the current study we don’t see the need to include these aspects. Nevertheless, identities of all protagonists was frequently verified by morphological characters. (see line 208)
12) Do not parasitoids or hyperparasitoids fall into the yellow traps or have they not been counted?
- see response above. Other parasitoid species were also attracted to yellow sticky traps and counted. In very low numbers Chalcidoids and a handful Ichneumonoids were counted. Moreover the specific role of aphid parasitoids in the context of aphid monitoring was topic of a follow up study and will be published elsewhere (see lines 514-517).
13) It would be convenient to explain in detail the state of development of the plants when the aphids are placed on the plants and to know how much time passes from when they are left on the plants until the traps are removed for the first time at week 25. This could clearly explain the massive capture of winged aphids in the first planting.
- most details can be found already in the material and method section. Missing details on trap placement were added (see lines 195-197)
Comments to the results
In figure 2 the controls could be linked to the results in compartments 1 and 2
In figures 3, 4 and 5 it would be better if the three forms studied on the plant (apterae, alatae and nymphs) and the winged ones captured in the trap were included in a single graph. In such a way that there would only be one figure that included experiments 1, 2 and control for the first experiment and another figure with the same structure for the second.
- this aspect was discussed in our working group already in detail. Since the graphs would then become very confusing from our point of view we prefer to have them separated. In addition, the y-axes are not on the same scale, so that some lines would no longer stand out strongly from the x-axis.
Comments to the discussion
It is going too far to think that the development of wings is due to the presence of parasitism, especially when the data on the release of parasitoids have not been discussed (or are not known).
- yes, you are right. Therefore, we highlight that presence of antagonists was unlikely to cause wing development in our study. (see lines 458-460)
The authors refer to general issues of the life cycle of Aphis gossypii, but it would be necessary to know how it behaves in the area in which the experiments are being carried out, because at the time when the tests are being carried out, Aphis gossypii may be producing winged for many months and present several peaks of production of winged or a constant production. Are there no studies on this aspect in those areas? There is also no data on catches with previous traps?
- yes, this in an important aspect. Unfortunately studies on the behaviour of Aphis gossypii in the field are not available for the campus area. Follow up studies in the same greenhouse at least will show data on trap catches in other years. Nevertheless, the migration aspect most likely varies also in different areas and should be definitely included when the aphid monitoring strategy will be implemented into practice.
For other comments and small errors, see also the attached corrected manuscript
- thank you very much for the additional hints, they were all considered in the revised version of the manuscript